# Genomic Profiling of Sarcomas: A Promising Weapon in the Therapeutic Arsenal

**DOI:** 10.3390/ijms232214227

**Published:** 2022-11-17

**Authors:** Raquel Lopes-Brás, Dolores Lopez-Presa, Miguel Esperança-Martins, Cecília Melo-Alvim, Lina Gallego, Luís Costa, Isabel Fernandes

**Affiliations:** 1Department of Medical Oncology, Hospital Santa Maria, Centro Hospitalar Universitário Lisboa Norte, 1649-028 Lisbon, Portugal; 2Department of Pathology, Hospital Santa Maria, Centro Hospitalar Universitário Lisboa Norte, 1649-028 Lisbon, Portugal; 3Sérgio Dias Lab, Instituto de Medicina Molecular, Faculdade de Medicina da Universidade de Lisboa, 1649-028 Lisbon, Portugal; 4Luís Costa Lab, Instituto de Medicina Molecular, Faculdade de Medicina da Universidade de Lisboa, 1649-028 Lisbon, Portugal

**Keywords:** cancer care, comprehensive genomic profiling, genomics, next-generation sequencing, rare tumor, sarcoma sequencing-directed therapy, targeted therapy

## Abstract

Sarcomas are rare malignant mesenchymal neoplasms, and the knowledge of tumor biology and genomics is scarce. Chemotherapy is the standard of care in advanced disease, with poor outcomes. Identifying actionable genomic alterations may offer effective salvage therapeutic options when previous lines have failed. Here, we report a retrospective cohort study of sarcoma patients followed at our center and submitted to comprehensive genomic profiling between January 2020 and June 2021. Thirty patients were included, most (96.7%) with reportable genomic alterations. The most common alterations were linked to cell cycle regulation (*TP53*, *CDKN2A/B*, and *RB1* deletions and *CDK4*, *MDM2*, and *MYC* amplifications). Most patients (96.7%) had microsatellite stability and low tumor mutational burden (≤10 muts/megabase (Mb); median 2 Muts/Mb). Two-thirds of patients had actionable mutations for targeted treatments, including five cases with alterations amenable to targeted therapies with clinical benefit within the patient’s tumor type, ten cases with targetable alterations with clinical benefit in other tumor types, and five cases with alterations amenable to targeting with drugs under investigation in a clinical trial setting. A significant proportion of cases in this study had actionable genomic alterations with available targeted drugs. Next-generation sequencing is a feasible option for identifying molecular drivers that can provide therapeutic options for individual patients. Molecular Tumor Boards should be implemented in the clinical practice to discuss genomic findings and inform clinically relevant targeted therapies.

## 1. Introduction

Sarcomas are a rare group of malignant neoplasms arising from connective tissue (mesenchymal cells) that include more than 100 different histological subtypes [1]. This group comprises less than 1% of malignant tumors in adults and has an estimated incidence of around 1.5 per 100,000 cases in Europe [2]. As rare tumors, the knowledge of their biology and genomic alterations remains scarce, although is slowly increasing over time. Sarcoma diagnosis relies, not only on morphological and immunohistochemical features, but also on molecular alterations, such as *EWS/FLI1* fusion in Ewing sarcoma (EWS), *SS18-SSX* fusion in synovial sarcoma, or *kit* mutation in gastrointestinal stromal tumors (GIST) [1]. From the molecular point of view, sarcomas are mainly categorized into two groups. The first comprises sarcomas with specific genetic alterations, such as chromosomal translocations resulting in fusion genes and specific oncogenic mutations [3,4]. The second includes sarcomas with complex karyotypes with multiple and non-specific genetic alterations that cannot be detected by karyotyping or in situ hybridization techniques. The detection of these aberrations can be accomplished by next-generation sequencing (NGS), a massively parallel sequencing technique that enables the simultaneous sequencing of millions of fragments per run.

From a therapeutic perspective, data on clinically actionable genomic alterations in sarcoma, including its prevalence and distribution among histological subtypes, is required to identify potentially effective treatment options [5]. The mainstay of treatment in sarcomas is *en bloc* surgery [6,7,8,9,10,11], with (neo-)adjuvant systemic treatment or radiotherapy also indicated in some cases. In unresectable or metastatic advanced disease, surgery may still be an option if radical treatment is feasible. However, curative resection is not an option in most cases, with treatment relying on systemic therapy and resulting in poor outcomes and dismal prognosis (5-year overall survival of 15%) [12]. Except for GIST and dermatofibrosarcoma protuberans (DFSP), in which systemic treatment is based on tyrosine kinase inhibitors (TKIs), for most other sarcomas it usually includes anthracycline-based chemotherapy (topoisomerase II inhibitor), ifosfamide (alkylating agent), docetaxel (microtubule-stabilizing agent), cisplatin (alkylating agent), methotrexate (anti-metabolite), trabectedin (alkylating agent), gemcitabine (nucleoside analog), or pazopanib [antiangiogenic TKI; vascular endothelial growth factor receptor and platelet-derived growth factor receptor (PDGFR) inhibitor]. Therefore, there is a clear and still unmet need for effective treatment options for advanced disease. Given this scenario, tumor genomic analysis may uncover molecular drivers capable of providing therapeutic alternatives in later lines of treatment. Additionally, genomic profiling may also be helpful in identifying alterations with prognostic but also predictive value that can be used to tailor the treatment strategy for each individual patient.

The aim of this study was to retrospectively identify genomic alterations in a cohort of sarcoma patients followed at a Portuguese sarcoma reference center and submitted to comprehensive genomic profiling (CGP) through FoundationOne^®^ Heme (FOH) testing.

## 2. Results

### 2.1. Clinicopathological Characteristics of the Study Cohort

A total of 38 sarcoma patients with FOH genomic testing performed in an 18-month period between January 2020 and June 2021 were identified. In 11 of these, the test could not be performed in the first sample assessed, and in eight, neither in the second sample. These eight-second failures included five cases (62.5%) of osteosarcoma (OS), two cases (25.0%) of well-differentiated liposarcoma (LPS), and one case (12.5%) of leiomyosarcoma (LMS). Regarding the remaining three cases, deoxyribonucleic acid (DNA) and ribonucleic acid (RNA) analysis was performed in the second sample for two, and DNA analysis only was performed in the third case, as a second sample was not available. This made up a final sample of 30 patients with available clinical data and genomic testing.

The study population had a median age at the time of sample collection of 55 (range, 17–79) years and comprised eight (26.7%) males (Table 1). The genomic test had reduced sensitivity in 12 samples (40.0%), due to their quality, and DNA analysis could only be performed in one sample (3.3%). Four patients (13.3%) had bone sarcoma (BS) and 27 (86.7%) had soft tissue sarcoma (STS). BS histology was EWS in two patients and OS and chondrosarcoma (CS) in one patient each. STS histological subtypes included seven LPS, five LMS, three STS not otherwise specified (NOS), two rhabdomyosarcomas (RMS; one RMS NOS, one alveolar RMS [ARMS] subtype), one soft tissue EWS, one extraskeletal myxoid CS (EMCS), one DFSP, one inflammatory myofibroblastic tumor (IMFT), one angiosarcoma, one GIST, one neurofibroma, one malignant peripheral nerve sheath tumor (MPNST), one synovial sarcoma, and one follicular dendritic cell sarcoma (FDCS) (Figure 1). All samples were formalin-fixed paraffin-embedded (FFPE) tissue specimens. Twenty-three samples (76.7%) were collected from primary tumors, of which 22 (95.7%) had never been exposed to treatment (either systemic or radiotherapy). Four samples (13.3%) concerned local recurrences, of which two (50%) had recurred following systemic therapy, and three samples concerned distant metastases after systemic therapy.

### 2.2. Genomic Alterations

Twenty-nine of the 30 patients (96.7%) included in the study had reportable genomic findings, accounting for a total of 108 molecular alterations, with an average of 3.6 molecular alterations per case (the loss of *CDKN2A/B* genes occurred simultaneously and was thus considered as one event; Figure 2). Sample median exon coverage ranged between 379x and 1033x. The most frequently altered genes were related to cell cycle regulation and RAS/RAF/MEK/ERK and PI3K/AKT signaling pathways. STS NOS, ARMS, and LPS were the sarcoma subtypes with the highest number of altered genes.

#### 2.2.1. Tumor Mutational Burden and Microsatellite Status

Tumor mutational burden (TMB) was determined in 29 patients (it could not be determined in one LMS) and was lower than 10 muts/megabase (Mb) in all but one (STS NOS). The median TMB was 2 muts/Mb (range, 0–16). A TMB of 16 muts/Mb was considered an actionable genomic finding, as immunotherapy could be proposed based on studies on other tumor types [13,14,15].

Microsatellite status was stable in all samples for which it could be determined. The exception was the referred STS NOS case, in which this status could not be determined with confidence. This patient also presented an MSH6 rearrangement in exon 1.

#### 2.2.2. Cell Cycle Regulation and TP53 Pathway

The most frequently altered genes in this study were related to cell cycle regulation, namely to the TP53 pathway (Figure 2). Globally, alterations in these genes accounted for 31.5% of the molecular events found. The most common alterations were in the *TP53* gene, which was altered in 40.0% of patients. The alterations identified comprised single nucleotide variants (SNV; *n* = 9, 8.3%; LPS, LMS, STS NOS, bone EWS), including missense and nonsense mutations, loss of exons (*n* = 4, 3.7%; ARMS, LPS, bone OS, bone CS), indels (*n* = 1, 0.9%; STS NOS), and rearrangements (*n* = 1, 0.9%; ARMS). In three cases, two alterations were found in this gene (ARMS with *TP53* rearrangement of exon 5 and loss of exons 2–4; LPS with P250S mutation and loss of exons 5–6; BEWS with R248Q and subclonal R273H mutations). Other altered genes implicated in cell cycle regulation included *CDKN2A/B* (gene loss, *n* = 10, 9.3%; MPNST, STS NOS, FDCS, bone OS, neurofibroma), *CDK4* (amplification, *n* = 3, 2.8%; LPS), *RB1* (splice site, *n* = 1, 0.9%; SNV, *n* = 1, 0.9%; STS NOS and LPS, respectively), *CCNE1* (amplification, *n* = 3, 2.8%; ARMS, STS NOS, and LMS), and *MDM2* (amplification, *n* = 2, 1.9%; LPS).

#### 2.2.3. RAS/RAF/MEK/ERK and PI3K/AKT Signaling Pathways

One (0.9%) KRAS SNV was found on an STS NOS sample. Regarding the PIK3/AKT pathway, *PIK3CA* alterations were found in two patients (6.7%), including a missense mutation in a RMS NOS and a multi-hit event in a bone CS. *PTEN* gene, a regulator of the PIK3/AKT pathway, was altered in five patients (16.7%) in the form of an indel in STS NOS, LMS, and bone CS, a nonsense mutation in angiosarcoma, and a loss of exon in EMCS. A *kit* indel was found in a patient with GIST (0.9%), and two *PDGFRA* events (1.9%) were also reported in the form of an indel in a patient with LPS and an amplification in a patient with bone CS.

#### 2.2.4. Fusions, Rearrangements, and Copy Number Alterations

Five fusion events (4.4%) were identified in this sarcoma cohort. In three cases, fusion involved the *EWRS1* gene: *EWSR1-FLI1* (type 2) and *EWSR1-FL1* (type 6/8) in EWS, and *EWSR1-NR4A3* in EMCS. The other two cases corresponded to a synovial sarcoma presenting the *SS18-SSX2* fusion and to an IMFT presenting the *CARS-ALK* fusion (Table 2).

Rearrangements were identified in *FANCA* (intron 32; LPS), *MEN 1* (exon 7; LPS), *MSH6* (exon 1; STS NOS), *NF1* (exon 38; MPNST), and *TP53* (exon 5; ARMS) (Table 2). Regarding copy number alterations (CNA), amplifications were observed in 15 (50.0%) patients (Table 3), and loss of exons/genes in 12 (40.0%) (Table 4). Apart from the previously mentioned genes, amplifications were also found in *BCL2L2* (1.8%; STS NOS and bone OS), *C17orf39* (1.8%; STS NOS and LMS), *CKS1B* (1.8%; two LMS), *FRS2* (1.9%; LPS), *ERBB2* (0.9%; MPNST), *MYC* (1.8%; angiosarcoma and ARMS), *CSF3R* (0.9%; ARMS), *ESR1* (0.9%; LPS), *HGF* (0,9%; STS NOS), *JUN* (0.9%; STS NOS), *KDM5A* (0.9%; bone OS), *MCL1* (0.9%, ARMS), MYC (1.9%; angiosarcoma and ARMS), *NTRK* (0.9%; ARMS), *RAD21* (0.9%; ARMS), and *RICTOR* (0.9%; STS NOS) genes. In addition to the loss of exons in *PTEN* and *TP53* and the loss of *CDKN2A/B* genes, loss of exons was also found in *ATRX* (0.9%; bone EWS), *BCOR* (0.9%; EMCS), *ETV6* (0.9%; angiosarcoma), *LRP1B* (0.9%; MPNST), and *NF1* (0.9%; MPNST). Loss of the *KDM6A* gene was identified in one (0.9%) patient with LPS.

#### 2.2.5. Actionability

Thirty-six alterations found in this study (31.9%, including in TMB) were actionable genomic drivers, representing a potential treatment opportunity with targeted therapies either approved (in the same or different tumor type) or under investigation in ongoing clinical trials (as basket trials; Figure 3; Table 5). Five (4.6%) molecular alterations were actionable drivers within the same tumor type, and 16 (14.8%) had a targeted therapy approved in other tumor types.

Twenty patients (66.7%) had at least one actionable genomic alteration. Of these, five had alterations with targeted therapies approved in the considered tumor type (although they could also present other findings amenable to targeted therapies approved in other cancers), 10 had alterations with targeted therapies approved in other tumor types, and five had alterations with not yet approved therapies and were only addressed in a clinical trial setting. Cases with alterations with therapies approved in the same tumor type included one GIST with *kit* in-frame insertion in exon 9 (proposed therapy: imatinib, regorafenib, sunitinib, avapritinib, ripretinib), one bone EWS with anaplastic lymphoma kinase (*ALK*) missense mutation (proposed therapy: entrectinib; of note, other ALK inhibitors, such as alectinib, brigatinib, and lorlatinib, were also proposed, although these TKIs are not approved in sarcomas), one IMFT with *ALK-CARS* fusion (also with entrectinib approved in sarcomas and other TKIs, namely brigatinib, ceritinib, crizotinib, alectinib, and lorlatinib, approved in other disease settings), one STS NOS with TMB of 16 mut/Mb (proposed therapy: pembrolizumab; immunotherapy with pembrolizumab is an agnostic indication in advanced solid tumors with high TMB [≥10 mut/Mb], according to the Food and Drug Administration [FDA]), and one STS CS with *EWSR1-NR4A3* fusion (proposed therapy: pazopanib).

At the time of data analysis, at least four patients (13.3%) had received directed therapy against a genomic target identified by NGS analysis (Table 6). One patient with GIST with a *kit* insertion received imatinib, and one patient with EMCS with an *EWSR1-NR4A3* fusion received pazopanib, both in the course of standard clinical practice and regardless of NGS results. The other two patients received sequencing-directed therapy after NGS analysis. One was a 62-year-old female with unresectable myxoid/round cell LPS and *PDGFRA* deletion, who received imatinib after progression on doxorubicin. Although the partial response was obtained in a CT scan one month after starting therapy, the patient was admitted to the hospital due to SARS-CoV2 pneumonia and died after two weeks. The second patient was a 28-year-old female with *PIK3CA*-mutated metastatic embryonal rhabdomyosarcoma, who was treated with everolimus after progression on vincristine, dactinomycin, and cyclophosphamide. A CT scan showed a partial response after two months of therapy, but less than one month later, the patient was admitted with a severe respiratory infection and died within one week. Both cases were discussed in a multidisciplinary meeting, and Molecular Tumor Board (MTB).

## 3. Discussion

Sarcomas are a very rare group of malignant neoplasms, with over one hundred histological subtypes and a multitude of possible cells of origin. Therefore, the understanding of their biology, clinical behavior, and genomic landscape is less robust than in other tumor types. Identifying therapeutic targets through genomic profiling has the potential to uncover new effective treatment options and thus improve clinical outcomes. This is particularly relevant in advanced disease, as curative treatment is not feasible and the outcomes with systemic therapy are disappointing. Being such a rare group of diseases, each subtype is extremely uncommon in the clinical practice, and thereby collaborative efforts are of utmost importance to fill the gaps in the knowledge of genomics and biology of these tumors.

The present sarcoma cohort is one of the largest Portuguese retrospective series of NGS genomic profiling using the RNA and DNA array platform FOH [16]. Its analysis revealed that the large majority (96.7%) of patients submitted to NGS had at least one genomic alteration, a TMB under 10 muts/Mb, and stable microsatellite status.

In one of the cases of STS NOS, TMB was 16 muts/Mb. This finding was considered an actionable event based on studies in other tumor types, as immune checkpoint inhibitors (ICI) could be proposed as a therapeutic option [13,15]. ICI target the immune checkpoint pathways that negatively regulate immune function against cancer cells [17]. These agents can block the cytotoxic T-lymphocyte–associated antigen 4 (CTLA-4)/B7 or the programmed death 1 (PD-1)/programmed cell death-ligand 1 (PD-L1), leading to T-cell mediated tumor cell destruction. CTLA-4 is primarily expressed by T cells and prevents the binding of CD28 to CD80/86, thereby blocking T cell proliferation signaling [18]. PD-L1 is overexpressed in tumor cells and binds to its receptor PD-1 on T cells, impairing their proliferation, differentiation, and activation [19]. PD-L1 levels in the tumor microenvironment have been used as a positive predictive biomarker for targeting the PD-1/PD-L1 axis with ICI. However, PD-L1 has shown to be an inconsistent biomarker of response to ICI, spurring the investigation of other molecules [20]. TMB represents an estimation of the tumor’s neoantigenic load [20]. Somatic mutations in tumor DNA have the potential to generate neoantigens, which can be recognized and targeted by T cells [21]. After transcription and translation, neopeptides can be processed and presented to T cells through the major histocompatibility complex. However, only a small minority of tumor DNA mutations can lead to neoantigens that are recognized by T cells. Therefore, the higher the number of somatic mutations, the more likely it is for neoantigens to be present. TMB varies across tumor types and the heterogeneous group of sarcoma neoplasms typically shows low levels of TMB [22]. As previously mentioned, FDA has approved pembrolizumab as an agnostic indication in advanced solid tumors with high TMB (≥10 mut/Mb). This was based on a multicenter single-arm phase II trial of various types of solid tumors (KEYNOTE-158). In patients with advanced solid tumors with high TMB (*n* = 102; including nine subtypes: anal, biliary, cervical, endometrial, mesothelioma, neuroendocrine, salivary, small-cell lung, thyroid, and vulvar), a clinically important objective response (29%; 95% confidence interval [CI] 21–39) was observed. In patients with low TMB, objective response was observed in 6% (CI 5–8). Following these findings, high TMB was considered an actionable genomic event for the use of ICI in solid tumors, although sarcomas were not included in this study.

The rate of first sample failures was 28.9%, and it mainly comprised bone sarcomas (four OS and one CS) and LPS (four LPS). The low success rate of CGP in bone sarcoma has been reported by others [23] and may be explained by the decalcification process, which may lead to subsequent DNA and RNA destruction and correlate with lower tumor cellularity and tissue volume.

The most frequently altered genes in this study were related to cell cycle regulation or RAS/RAF/MEK/ERK and PI3K/AKT signaling pathways. As reported in other studies [24,25,26,27,28,29], TP53 was the gene most frequently presenting genomic alterations (40.0% of samples). Other frequently altered genes were *CDKN2A/B* (16.7%), *PTEN* (16.7%), and *ATRX* (13%). These findings are in line with other publications, which have additionally reported high rates of molecular alterations in the *RB1* gene, which was not observed in the present study [25,26]. Regarding gene fusions, these were found in five patients in this study. The *EWRS1* fusion corresponded to Ewing sarcoma (*FL1* gene) and EMCS (*NR4A3* gene), and the *SS18-SSX* fusion corresponded to synovial sarcoma, as would be expected [1,30,31,32]. More than being acknowledged therapeutic targets, these genomic events currently have a diagnostic value, being under assessment in clinical trials. Concerning CNA, amplifications were found in half of the patients, and loss of genes/exons in 40%. Gene amplifications are frequently present in sarcomas and were also identified in this study, namely *CDK4* and *MDM2* in 10.0% and 7.0% of cases, respectively. The *MDM2* amplification occurred exclusively in the presence of the *CDK4* amplification and corresponded to two cases of LPS, a finding that had also been reported in this subtype by Goirsberg et al. and Thway et al. [25,33].

Most patients in this cohort (*n* = 21; 70.0%) had at least one actionable molecular alteration, although it was a confirmed target for therapies with proven clinical benefit within the considered tumor type only in a minority of them (*n* = 5; 16.7%). This highlights the great unmet need for targeted therapies in sarcoma. Moreover, this group of patients already had an indication for specific gene testing (i.e., *kit* in GIST) or a drug indication as per disease type (i.e., pazopanib in chondrosarcoma). A considerable number of patients (*n* = 15; 50.0%) had alterations targetable by therapies without demonstrated clinical benefit within the considered tumor type. These patients probably represent those for whom the benefit of comprehensive NGS is most valuable. CGP could reveal new targets not yet explored in sarcoma and potentially amenable to targeting with specific therapies with proven results in other cancers or in preclinical/early-phase of development in sarcoma clinical trials. It is also worth mentioning that, although CGP is a powerful tool to detect molecular drivers, broader sequencing setups, such as whole-exome sequencing or whole-genome sequencing, could further expand the detection of mutations potentially candidates for therapeutic targeting or biomarkers in clinical trials [34]. MTBs are extremely important to interpret and discuss genomic findings and inform clinically relevant sequencing-directed therapies. Moreover, with the ever-growing knowledge of cancer genomics, the MTB also plays a relevant role in the journey toward personalized cancer care. Therefore, its implementation in Oncology departments should go hand in hand with the expanding use of CGP in clinical practice [35].

Limitations of our study include the small sample size, the heterogeneity of the sample, the retrospective nature of this study, and the fact that some sarcomas have an identified specific driver mutation. First, sarcomas are a rare group of tumors as previously mentioned and this contributes to the small sample, as well as the fact that CGP is not reimbursed by the Public Health System in Portugal, leading to a smaller percentage of sarcoma patients taking the genomic test. Second, since this is a retrospective study, heterogeneity of the tumor types included is expected and adds to the complex analysis of genomic findings in our research, as it limits subgroup analysis. Other studies have overcome this sample heterogeneity by doing multicentric/network centres prospective studies [26,36]. Finally, as a group of sarcomas have specific single-driver molecular events, it is difficult to interpret the role of other genomic findings, as the carcinogenic effect of the pathognomonic molecular alteration may be dominant over other genomic events (passenger mutations).

## 4. Materials and Methods

Soft tissue and bone sarcoma patients at any stage of disease with FFPE tissue samples submitted to FOH genomic testing and with test results available during standard clinical care between January 2020 and June 2021 were included in the analysis. All tumor samples were assessed by a Pathology expert in sarcomas. CGP by FOH testing included both DNA sequencing of 406 cancer-related genes and RNA sequencing of 265 commonly rearranged genes in cancer (Appendix A) [37]. The accuracy of the FOH test can be found in the Appendix A [37]. The analysis of genomic alterations included base substitutions, indels, amplifications, copy number alterations, and gene fusions/rearrangements, TMB (reported as muts/Mb), and microsatellite status. TMB is determined by measuring the number of somatic mutations in sequenced genes on the FOH test and extrapolating to the genome as a whole [37]. Microsatellite status, which is a measure of microsatellite instability, is determined by assessing indel characteristics at 114 homopolymer repeat loci in or near the targeted gene regions of the FOH test [37]. Actionable genomic alterations were classified in the FOH report as associated with an approved targeted therapy (in the patient’s tumor type or other) or with an investigational targeted therapy. Clinical data were retrieved from patients’ clinical registries. Informed consent for genomic testing was obtained as per standard practice. Ethical approval for this study, including a waiver of informed consent, was provided by the Institutional Review Board of Centro Académico de Medicina de Lisboa (project approval number 198/21).

## 5. Conclusions

This study represents one of the largest in Portugal characterizing molecular alterations in sarcoma through DNA and RNA NGS. Overall, its findings add to the previously reported pool of genomic findings in sarcoma, a rare disease with a paucity of approved therapies. The study highlights the importance of NGS testing in clinical practice as a tool to uncover new therapeutic avenues in a disease with very limited options and mainly relying on chemotherapy in the therapeutic armamentarium, with poor outcomes. The growing use of genomic profiling should go side by side with the implementation and widespread adoption of MTBs in the Oncology clinical practice.

## Figures and Tables

**Figure 1 ijms-23-14227-f001:**
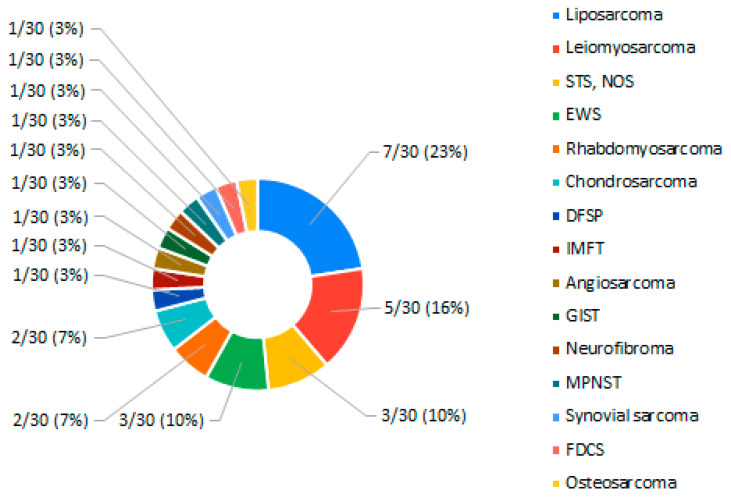
Distribution of sarcoma subtypes in the study population. DFSP—dermatofibrosarcoma protuberans; EWS—Ewing sarcoma; FDCS—follicular dendritic cell sarcoma; GIST—gastrointestinal stromal tumor; IMFT—inflammatory myofibroblastic tumor; MPNST—malignant peripheral nerve sheath tumor; NOS—not otherwise specified; STS—soft tissue sarcoma.

**Figure 2 ijms-23-14227-f002:**
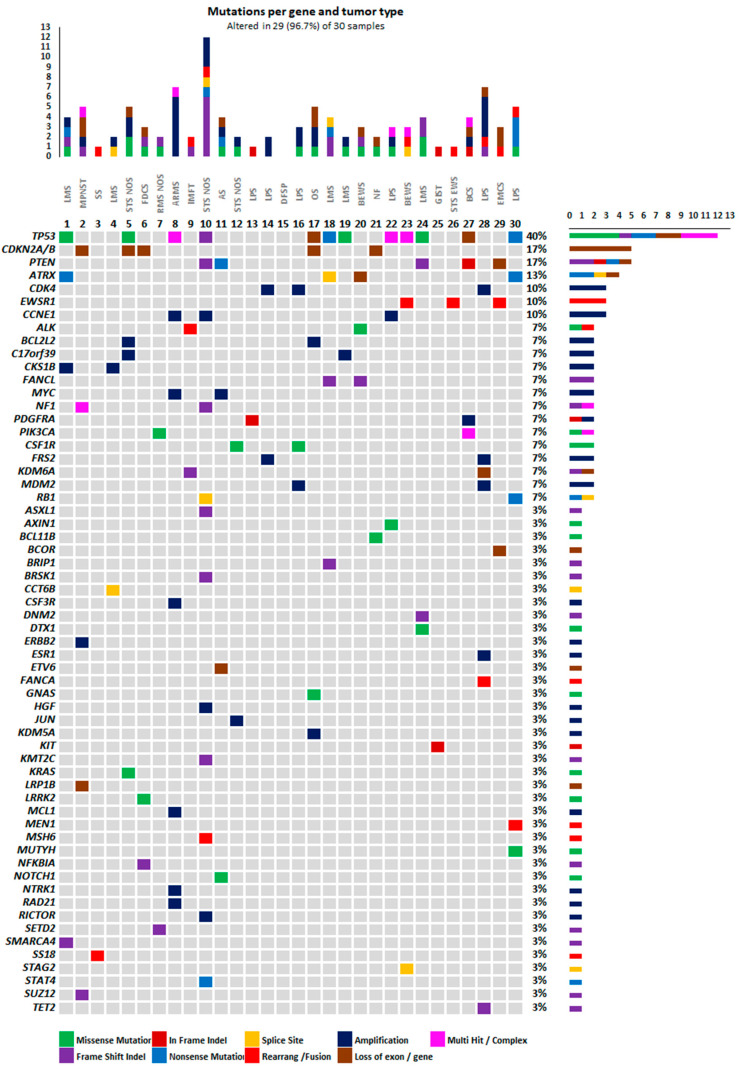
Waterfall plot of molecular alterations identified in the study population. ARMS—alveolar rhabdomyosarcoma; AS—angiosarcoma; BCS—bone chondrosarcoma; BEWS—bone Ewing sarcoma; DFSP—dermatofibrosarcoma protuberans; EMCS—extraskeletal myxoid chondrosarcoma; EWS—Ewing sarcoma; FDCS—follicular dendritic cell sarcoma; GIST—gastrointestinal stroma tumor; IMFT—inflammatory myofibroblastic tumor; LMS—leiomyosarcoma; LPS—liposarcoma; MPNST—malignant peripheral nerve sheath sarcoma; NF—neurofibroma; NOS—not otherwise specified; OS—osteosarcoma; RMS—rhabdomyosarcoma; SS—synovial sarcoma; STS—soft tissue sarcoma.

**Figure 3 ijms-23-14227-f003:**
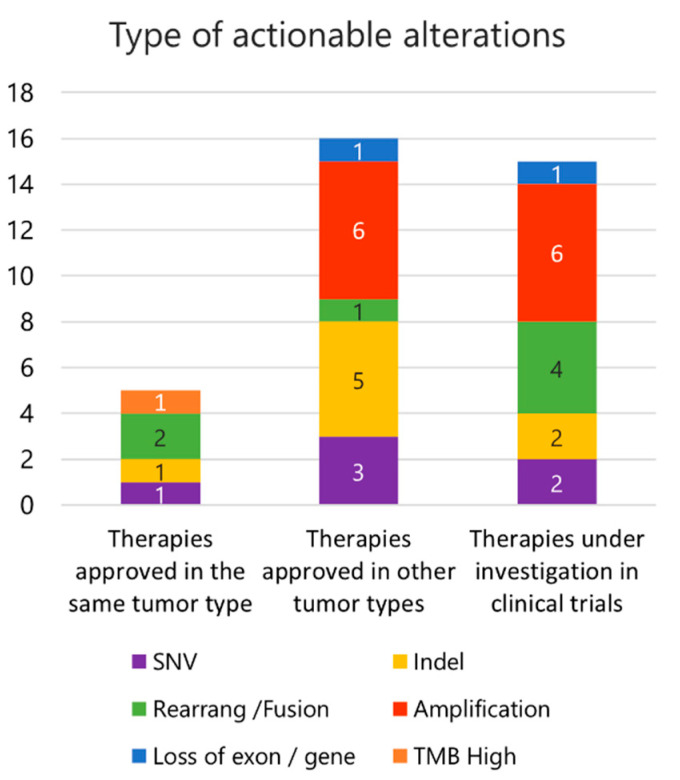
Genomic alterations with approved therapies in the study cohort, by type of alteration.

**Table 1 ijms-23-14227-t001:** Clinicopathological characteristics of the study population.

Clinicopathological Characteristic	(*n* = 30)	
Age | median (range)	55 (17–79)	
Female gender | n (%)	22 (73.3)	
Sensitivity—reduced due to sample quality | n (%)	12 (40.0)	
DNA and RNA analysis | n (%)	29 (96.7)	
Sample collection location | n (%)	Treatment-naïve	Previous systemic treatment
Primary tumour	1 (3.3)	22 (73.3)
Local recurrence	2 (6.7)	2 (6.7)
Distant metastasis	0 (0.0)	3 (10.0)

DNA—deoxyribonucleic acid; RNA—ribonucleic acid.

**Table 2 ijms-23-14227-t002:** Fusions and rearrangements identified by FoundationOne^®^ Heme testing.

Genomic Event	Sample Nr	Pathology	Gene	Genomic Finding
Fusion	3	SS	*SS18*	*SS18-SSX2* fusion
	9	IMFT	*ALK*	*CARS-ALK* fusion
	23	BEWS	*EWSR1*	*EWSR1-FLI1* fusion (type 8/6)
	26	STS EWS	*EWSR1*	*EWSR1-FLI1* fusion (type 2)
	29	EMCS	*EWSR1*	*EWSR1-NR4A3* fusion
Rearrangement	2	MPNST	*NF1*	*NF1* rearrangement exon 38
	8	ARMS	*TP53*	*TP53* rearrangement exon 5
	10	STS NOS	*MSH6*	*MSH6* rearrangement exon 1
	28	LPS	*FANCA*	*FANCA* rearrangement intron 32
	30	LPS	*MEN1*	MEN1 rearrangement exon 7

ARMS—alveolar rhabdomyosarcoma; BEWS—bone Ewing sarcoma; EMCS—extraskeletal myxoid chondrosarcoma; EWS—Ewing sarcoma; IMFT—inflammatory myofibroblastic tumor; LPS—liposarcoma; MPNST—malignant peripheral nerve sheath sarcoma; NOS—not otherwise specified; OS—osteosarcoma; SS—synovial sarcoma; STS—soft tissue sarcoma.

**Table 3 ijms-23-14227-t003:** Amplifications identified by FoundationOne^®^ Heme testing.

Sample Nr	Pathology	Amplified Genes
1	LMS	*CKS1B*
2	MPNST	*ERBB2*
4	LMS	*CKS1B*
5	STS NOS	*BCL2L2, C17orf39*
8	ARMS	*MYC*, *CCNE1*, *CSF3R*, *MCL1*, *NTRK1*, *RAD21*
10	STS NOS	*HGF*, *RICTOR*, *CCNE1*
11	AS	*MYC*
12	STS NOS	*JUN*
14	LPS	*CDK4*, *FRS2*
16	LPS	*CDK4*, *MDM2*
17	OS	*BCL2L2*, *KDM5A*
19	LMS	*C17orf39*
22	LPS	*CCNE1*
27	BCS	*PDGFRA*, *PIK3CA*
28	LPS	*CDK4*, *MDM2*, *ESR1*, *FRS2*

ARMS—alveolar rhabdomyosarcoma; AS—angiosarcoma; BCS—bone chondrosarcoma; LMS—leiomyosarcoma; LPS—liposarcoma; MPNST—malignant peripheral nerve sheath sarcoma; NOS—not otherwise specified; OS—osteosarcoma; STS—soft tissue sarcoma.

**Table 4 ijms-23-14227-t004:** Loss of genes or exons identified by FoundationOne^®^ Heme testing.

Sample Nr	Pathology	Genomic Finding
2	MPNST	*NF1*—loss of exons 1–38; *CDKN2A/B* loss
5	STS NOS	*CDKN2A/B* loss
6	FDCS	*CDKN2A/B* loss
8	ARMS	*TP53*—loss of exons 2–4
11	AS	*ETV6*—loss of exons 2–5
17	OS	*CDKN2A/B* loss; *TP53*—loss of exons 1–9
20	BEWS	*ATRX*—loss exons 2–9
21	NF	*CDKN2A/B* loss
22	LPS	*TP53*—loss of exons 5–6
27	BCS	*TP53*—loss of exons 8–9
28	LPS	*KDM6A* loss
29	EMCS	*PTEN*—loss of exons 1–4; *BCOR* loss

ARMS—alveolar rhabdomyosarcoma; AS—angiosarcoma; BCS—bone chondrosarcoma; BEWS—bone Ewing sarcoma; EMCS—extraskeletal myxoid chondrosarcoma; FDCS—follicular dendritic cell sarcoma; LPS—liposarcoma; MPNST—malignant peripheral nerve sheath sarcoma; NF—neurofibroma; NOS—not otherwise specified; OS—osteosarcoma; STS—soft tissue sarcoma.

**Table 5 ijms-23-14227-t005:** Actionable genomic alterations and respective therapeutic options.

Sample	Diagnosis	Genomic Finding	Therapies with Clinical Benefits within Patient’s Tumor Type	Therapies with Clinical Benefits in Other Tumor Type	Nr of Available Clinical Trials
2	MPNST	*ERBB2*—amplification	none	Ado-trastuzumab emtansine Afatinib Dacomitinib Fam-trastuzumab deruxtecan Lapatinib Neratinib Pertuzumab Trastuzumab	10 trials
		*NF1*—rearrangement exon 38, loss exons 1–38	none	Binimetinib Cobimetinib Selumetinib Trametinib	10 trials
5	STS NOS	*KRAS*—Q61L	none	none	9 trials
7	RMS NOS	*PIK3CA*—N345I	none	Everolimus Temsirolimus	10 trials
8	ARMS	*MYC*—amplification	none	none	6 trials
9	IMFT	*ALK*—*CARS-ALK* fusion	none	Brigatinib 2A Ceritinib 2A Crizotinib 2A Alectinib Lorlatinib	8 trials
10	STS NOS	Tumor Mutational Burden—16 Muts/Mb	Pembrolizumab	Atezolizumab Avelumab Cemiplimab Durvalumab Nivolumab Nivolumab + Ipilimumab	10 trials
		Microsatellite status—Cannot Be Determined	none	none	None
		*NF1*—Y628fs*3	none	Selumetinib Trametinib	10 trials
		*PTEN*—N63fs*36	none	Everolimus Temsirolimus	10 trials
		*HGF*—amplification	none	none	4 trials
		*RICTOR*—amplification	none	none	6 trials
11	AS	*PTEN*—W274*	none	Everolimus Temsirolimus	10 trials
		*MYC*—amplification	none	none	5 trials
13	LPS	*PDGFRA*—R841_D842del	none	Imatinib Sorafenib	7 trials
14	LPS	*CDK4*—amplification	none	Abemaciclib	10 trials
16	LPS	*CDK4*—amplification	none	Abemaciclib	10 trials
		*MDM2*—amplification	none	none	4 trials
18	LMS	*BRIP1*—N576fs*2	none	Niraparib Olaparib Rucaparib Talazoparib	10 trials
		*FANCL*—S176fs*8	none	none	10 trials
20	BEWS	*ALK*—F1174C	Entrectinib	Alectinib Brigatinib Lorlatinib	2 trials
		*FANCL*—S351fs*2	none	none	10 trials
23	BEWS	*EWSR1*—*EWSR1-FLI1* fusion (type 8/6)	none	none	3 trials
24	LMS	*PTEN*—N323fs*2	none	Everolimus Temsirolimus	10 trials
25	GIST	*KIT*—Y503_F504insAY	Imatinib Regorafenib Sunitinib Avapritinib Ripretinib	Nilotinib Sorafenib Dasatinib Ponatinib	10 trials
26	STS EWS	*EWSR1*—*EWSR1-FLI1* fusion (type 2)	none	none	5 trials
27	BCS	*PIK3CA*—amplification, R93W	none	Everolimus Temsirolimus	10 trials
		*PDGFRA*—amplification	none	Imatinib	1 trial
		*PTEN*—Y178del	none	none	10 trials
28	LPS	*CDK4*—amplification	none	Abemaciclib	10 trials
		*FANCA*—rearrangement intron 32	none	none	10 trials
		*MDM2*—amplification	none	none	4 trials
29	EMCS	*EWSR1*—*EWSR1-NR4A3* fusion	Pazopanib	Sunitinib	3 trials
		*PTEN*—loss exons 1–4	none	none	10 trials
30	LPS	*MEN1*—rearrangement exon 7	none	none	10 trials

ARMS—alveolar rhabdomyosarcoma; AS—angiosarcoma; BCS—bone chondrosarcoma; BEWS—bone Ewing sarcoma; EMCS—extraskeletal myxoid chondrosarcoma; EWS—Ewing sarcoma; GIST—gastrointestinal stroma tumor; IMFT—inflammatory myofibroblastic tumor; LMS—leiomyosarcoma; LPS—liposarcoma; MPNST—malignant peripheral nerve sheath sarcoma; NOS—not otherwise specified; RMS—rhabdomyosarcoma; STS—soft tissue sarcoma.

**Table 6 ijms-23-14227-t006:** Patients treated with targeted therapies in accordance with NGS results.

Sample	Age	Histopahological Diagnosis	Title 3	Title 4
7	28	RMS NOS	*PIK3CA*—N345I	CAV ^1^ × 4 cycles → DP Everolimus 10 mg × 3 cycles → PR → DP OS ^2^ 4 months
13	61	LPS	*PDGFRA*—R841_D842del	Doxorubicin 75 mg/m^2^ × 3 cycles → DP Imatinib 400 mg × 2 cycles → PR → DP OS ^2^ 2 months
25	42	GIST	*KIT*—Y503_F504insAY	Imatinib 400 mg × 4 → DP Imatinib 600 mg × 3 cycles → DP Imatinib 800 mg × 34 cycles → SD → DP Sunitinib 50 mg × 6 cycles → DP Regorafenib 160 mg × 9 cycles → SD → DP Rechallenge Imatinib 400 mg × 16 cycles → SD → DP Ripretinib 150 mg × 9 cycles → SD → DP Ripretininb 300 mg × 3 cycles → DP OS ^2^ 95 months
29	43	EMCS	*EWSR1*—EWSR1-NR4A3 fusion	Pazopanib 800 mg × 7 cycles → DP Doxorubicin 75 mg/m^2^ × 3 cycles → DP Gemcitabine 1000 mg/m^2^ × 3 cycles → DP Trabectedin 1.5 mg/m^2^ × 4 cycles DP OS ^2^ 18 months

^1^ Actinomycin D 0.75 mg/m^2^; Vincristine 2 mg/m^2^; Cyclophosphamide 1200 mg/m^2^. ^2^ Since the beginning of targeted therapy. DP—disease progression; EMCS—extraskeletal mixoyd chondrosarcoma; LPS—liposarcoma; GIST—gastrointestinal stromal tumor; OS—overall survival; PR—partial response; RMS NOS—rhabdomyosarcoma not otherwise specified; SD—stable disease.

## Data Availability

Data sharing is not applicable to this article.

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
