# Peer review of "Genomic Profiling of Sarcomas: A Promising Weapon in the Therapeutic Arsenal"

_ijms, 2022, doi:10.3390/ijms232214227_

Round 1

Reviewer 1 Report

This manuscript reports the molecular profiling of 30 cases of sarcoma using a panel of Roche FoundationOne Heme. The most frequently mutated were TP53, CDKN2A/B, PTEN, and ATRX.

The research used a commercially available panel from Roche (Foundation Medicine). The technical details of this panel are partially available. Some information is available in a manuscript, and in a webpage link.

 He J et al. Integrated genomic DNA/RNA profiling of hematologic malignancies in the clinical setting. Blood. 2016 Jun 16;127(24):3004–14. doi: 10.1182/blood-2015-08-664649. Epub 2016 Mar 10. PMID: 26966091; PMCID: PMC4968346.

https://www.foundationmedicine.qarad.eifu.online/foundationmedicine/en/foundationmedicine?keycode=454269579

 The manuscript is well written, and it is easy to read and understand. Tables summarizing the information in the text would help ease reading. The manuscript could be improved if more detailed information is provided about the clinicopathological characteristics of the patients, and the technical aspects of the NGS analyses.

Additional comments:

 (1) Lines 62–63. The following drugs are written: “anthracycline-based chemotherapy, ifosfamide, docetaxel, cisplatin, methotrexate, trabectedin, gemcitabine, or pazopanib.” Could you please state the mechanism of action of each drug?

 (2) Could you please add a list of acronyms (abbreviations) at the end of the manuscript?

 (3) Lines 84–98. I found this section a little difficult to follow. Could you please summarize the information in one table? E.g., type of sarcoma, number of cases, diagnostic or relapse biopsy, age, location, treatment.

 (4) Figure 1. Could you please provide the number of cases in each sarcoma type as **/30 (%).

 (5) Lines 129–131. Regarding “A TMB of 16 muts/Mb was considered an actionable genomic finding, as immunotherapy could be proposed based on studies on other tumor types [14-16].” Could you please provide more information? Why immunotherapy depends on the mutation burden? (PD-L1, CTLA-4, PD-1?)

 (6) Usually, names of genes are written in italics.

 (7) Regarding section 2.2.4. “Fusions, rearrangements, and copy number alterations.” The second paragraph is dense. Have you considered making a table instead of writing all the details?

 (8) Regarding section 2.2.5. “Actionability.” The type of mutation (actionable genomic driver), type of sarcoma, and the specific drug could be summarized in a table.

 (9) Line 236. Regarding “NGS genomic profiling using the RNA and DNA array platform FoundationOneHeme.” Could you please add the webpage of this company?

 (10) Regarding the methods. In its current form, if available, more information could be added.

(A) As I understand, this research used the “FoundationOneHeme” test from Foundation Medicine (Roche).

Webpage: https://www.rochefoundationmedicine.com/home/services/heme.html

Sample report (with list of genes): https://rochefoundationmedicine.com/F1Hemereport_AML

(B) Could you please add as an appendix or supplementary file the DNA gene list for detecting base substitutions, insertions/deletions, and copy number alteration, the DNA gene list for rearrangements, and the RNA list for rearrangements?

(C) How was the microsatellite status (a measure of microsatellite instability (MSI)) assessed?

(D) Was the tumor mutation burden extrapolation to the genome as a whole?

(E) What was the accuracy of the test?

(F) What was the median exon coverage for the samples?

(G) Was the DNA extracted from FFPET?

 (11) Regarding the material. Could you please provide the clinicopathological characteristics of the samples? Age, sex, histological characteristics, treatment, response to treatment and survival?

(12) Sarcomas group include many types. Therefore, since 30 cases were studied, could you please comment about the limitations of this research?

Reviewer 2 Report

This study is important for planning the management of sarcomas. However, a larger study population would be desirable in the future. Also, prospective studies on fresh specimens would be helpful to identify mutations and decrease the risk of specimen loss.

Please rewrite this unclear sentence lines 63 through 65: "Given this scenario, tumor genomic analysis may uncover molecular drivers capable of providing therapeutic (although mostly off-label) options for later lines, where there is a clear and still unmet need for treatment alternatives."

Reviewer 3 Report

In this manuscript, the authors reported their findings on 30 cases of sarcomas by comprehensive genomic profiling. Sarcomas are a class of rare malignant mesenchymal neoplasms, and identifying actionable genomic alterations may provide clues for targeted therapies for those patients. Though the sample size is not that large, the data will add values on reported pool of genomic findings in sarcoma. I would suggest that the authors can discuss the limitations of their study by comparing with existing studies like https://www.cell.com/cell/fulltext/S0092-8674(17)31203-5
